# Using Petrogeochemical Modeling to Understand the Relationship between Paleozoic Magmatism in the Kola Region and Its Precambrian History

**Nikolay E. Kozlov [1,\*], Nikolay O. Sorokhtin [2] and Eugeny V. Martynov [1]**

[1]   Geological Institute of the Kola Science Centre RAS, 14 Fersman Street, 184209 Apatity, Russia; mart@geoksc.apatity.ru

[2]   P.P. Shirshov Institute of Oceanology RAS, 36 Nakhimov Prospect, 117997 Moscow, Russia; nsorokhtin@mail.ru

\*   Correspondence: kozlov@geoksc.apatity.ru; Tel.: +7-81555-79-656

**Abstract:** The Kola region hosts numerous Paleozoic massifs of ultrabasic alkaline rocks and carbonatites with deposits of commercially valuable metals, such as iron, tantalum, niobium, and rare earth elements. These magmatic complexes are characterized by high contents of alkaline elements at generally low contents of $SiO_2$ and/or $Al_2O_3$. In this study, we examined the precursors to the formation of the unique Paleozoic alkaline province through studying the early Precambrian stages in the evolution of the Kola collision area, from where these unique features probably originated. We mathematically modeled the changes in the chemical composition of these rocks. The obtained data can be used for metallogenic forecasting, which indicated a number of Precambrian objects in the region, namely, the Lapland Granulite Belt of the Kola region and granulite belts in Eurasia. The mathematical modeling performed during this research depicted a linear trend that defined the style of the changes in the chemical composition at the transition from the metaultrabasic-basic rocks of the Lapland granulite belt to the group of belts in Eurasia. These differences are statistically significant with respect to the obtained trend (chemical composition projected on the trend), mainly manifested as increased $SiO_2$ and $Al_2O_3$ contents with a decreasing total alkalis content, which is opposite to the indicated trends of the changing chemical composition in the Paleozoic alkaline rock units of the Kola region. We concluded that one of the reasons for the unique composition of the Paleozoic magmatism products could be a specific feature of the earlier Neoarchean stages of the tectonic-magmatic activity in the northeastern Baltic Shield, which implies a close relationship between later geological events and the early Precambrian history, at least in the study area.

**Keywords:** Kola region; alkaline province; Precambrian; ancient volcanogenic complexes

## 1. Introduction

The Kola region ranks in the leading position among the Precambrian provinces with its giant and high-grade mineral reserves of apatite, apatite-magnetite, phlogopite, and vermiculite deposits; rare metal mineralization (yttrium, firstly); and platinum group element (PGE) ores. The researchers who studied this province in detail considered it unique and one of the largest alkaline provinces in the world in terms of the scope of its multi-stage alkaline magmatism [1–5]. These and other authors studied the genesis of alkaline complexes mainly within the framework of the Paleoproterozoic stage in the magmatic activation of the region, with no reference to its prehistory. An earlier and possible relationship between the Paleoproterozoic platinum-group metal profile of the Kola region and the specific nature of Archean magmatism was reported [6]. In this study, we continued this research by

reviewing the precursors of the unique Paleozoic alkaline province formation by studying early stages in the evolution of the Kola collision area, from where such unique features could have originated. Our research is based on the results of the mathematical modeling of the patterns of change in the chemical rock composition represented by the contents of major element oxides ($SiO_2$, $TiO_2$, $Al_2O_3$, $\sum Fe = FeO + 0.9 \times Fe_2O_3$, $MgO$, $CaO$, $Na_2O$, and $K_2O$) at the transition from the rocks of the Lapland granulite belt to the other granulite belts of Eurasia. Our findings do not settle this issue for all specific rock assemblages and related mineral resources; we only address the possible relationship between the processes related to the formation of the Kola alkaline province and the Precambrian history of the regional evolution. Mathematical modeling was applied to search for an optimal description (in the form of a linear trend) of the mechanisms that changed the chemical composition of the rocks from the Lapland Granulite Belt to other similar belts in Eurasia. The results of this research can be valuable for specialists engaged in metallogenic analysis.

## 2. Geological Setting

The Kola alkaline province is located in the northeastern part of the Baltic Shield, and hosts the world's largest plutons of agpaitic syenites and numerous intrusions of alkaline ultrabasic rocks with carbonatites (Figure 1).

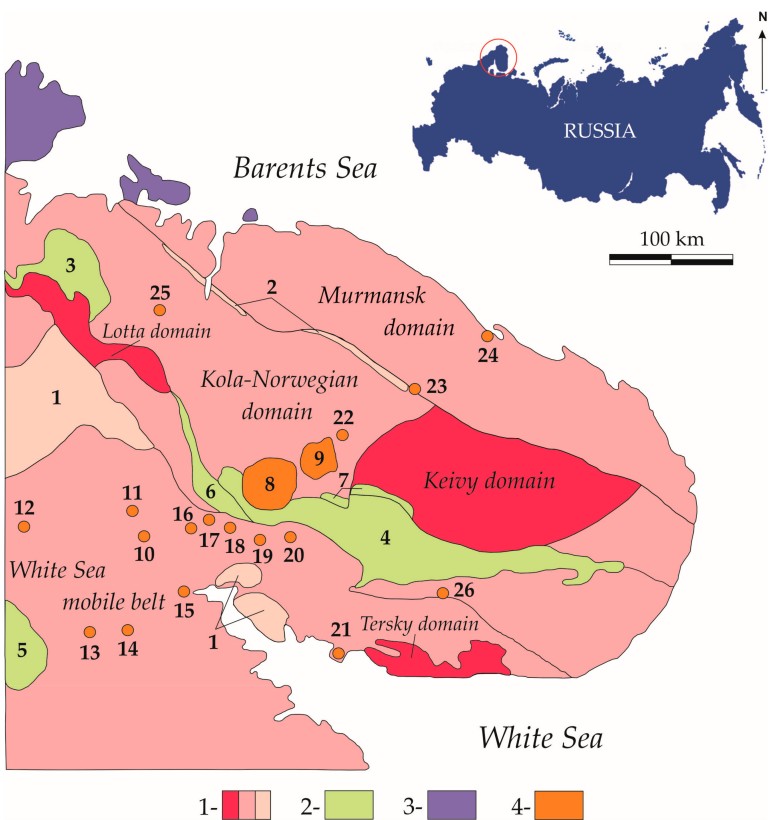

**Figure 1.** Scheme of the location of the geological sites with mantle magmatism in the Kola region (modified from Arzamastsev et al., [7]). Different colors indicate the structural and compositional complexes in the following: (1) Archean, (2) Paleoproterozoic, (3) Riphean, and (4) Paleozoic. Numbers indicate the following: Archean belts: (1) Lapland Granulite Belt, (2) Kolmozero–Voronya and area, Paleoproterozoic complexes: (3) Pechenga, (4) Imandra–Varzuga, (5) Pana–Kuolayarvi, (6) Monchetundra massif, (7) Fedorov–Pana Tundra massif; Paleozoic intrusions: (8) Khibiny, (9) Lovozero, (10) Niva, (11) Mavraguba, (12) Kovdor, (13) Sallanlatva, (14) Vuoriyarvi, (15) Kandaguba, (16) Afrikanda, (17) Ozernaya Varaka, (18) Lesnaya Varaka, (19) Salmagora, (20) Ingozero, (21) Turiy Mys, (22) Kurga, (23), Kontozero, (24) Ivanovka, (25) Sebl´yavr, and (26) Pesochny.

Paleozoic volcanic formations are outcroppings in the caldera depressions of the Kontozero, Lovozero, Khibiny, and Ivanovsky massifs [3], dike swarms, and explosion pipes. Rb-Sr dating was used to estimate the age of this magmatism as 380–360 million years [8]. This suggests that they were formed in a single stage of the tectonomagmatic activation [2].

We studied this stage in association with the preceding mantle magmatism, which had manifested in the Kola region before that in two phases [6], as follows:

(1)     Neoarchean (Neoarchean–Paleoproterozoic according to some data): emplacement of the Lapland Granulite and Kolmozero–Voronya belts (Figure 1); and

(2)     Paleoproterozoic: emplacement of basic-ultrabasic massifs, such as the Monchetundra and Fedorov–Pana intrusion, as well as the Imandra–Varzuga, Pechenga, and Pana–Kuolayarvi protorifts (Figure 1).

Thus, the Paleozoic stage was preceded by a long-term amagmatic period of the Baltic Shield evolution that lasted over 1.3 Ga [2]. Notably, no evidence exists of later magmatism in the region.

## 3. Materials and Methods

Our study was based on the previously applied approach of comparing rock assemblages that refer to the same type in the petrogeochemical nomenclature. The ultrabasic-basic rocks, of which metamorphosed analogues are identified the most reliably, were chosen as the study objects. Data on the composition of these rocks, where the genesis is associated with the mantle, discloses some of its geochemical characteristics. Our studies were based on the following provisions, two of which are justified in the above-mentioned publications. The chemical composition of the ultrabasic-basic rocks of the oceanic structures differ from that of the continental rocks throughout the Earth's geological history, since the Archean. This allows for assuming similar features of the mantle composition in geodynamically similar zones [9].

On the basis of the general notions on the geological structure of the region [6], it is possible to assume that since the Neoarchean, at least, the Earth's crust within the Kola region has not significantly shifted regarding its deep-seated components. Thus, the objective of this study was set with consideration that the petrological and geochemical features of the structural-compositional complexes produced by the Paleozoic mantle might have inherited these features of the mantle at earlier stages of the geological evolution of the region.

We relied on published data [9] on the chemical composition of ultrabasic–basic rocks from the high-pressure granulite belts of Eurasia located on the Baltic Shield (northeastern part), Anabar Shield, Aldan Shield (southern and southwestern parts), Ukranian Shield (Pobuzhye) in North East Asia; South and Southeast India; and North China. In total, about 850 bulk rock chemical analyses were selected for examination. Figure 2 indicates the location of the study objects. For the purpose of comparison, we selected rocks that are similar in terms of their primary nature and have protoliths, which may be interpreted as picrites and basic rocks. Today, these are represented by garnet and garnet-free amphibolites, pyroxene-bearing amphibolites, hypersthene, hypersthene–garnet, biotite–garnet–hypersthene, garnet–hypersthene, and garnet–bipyroxene crystalline schists in all of the studied rock units. As a result of the large volume of data used in the research, showing all of the sampling points on the map was impossible. We note that all of the studied rock units evenly covered the sampling grid. More details about this are available [9], where evidence was provided of the similar geological structure and chemical composition of these rock units, which were reconstructed as ancient island arcs, with mainly preserved initial features.

These objects were reconstructed as ancient island arcs, which mostly preserved their own original features [9]. The aim of our research was not to discuss this issue in detail, or to provide evidence of the isochemical metamorphism nature of the studied rock assemblages. Notably, the possible preservation of the relic protolith features of the chemical composition has been reported for all of these objects,

as well as the principal possibility that the rock compositions were preserved under the regional metamorphism setting in general [9].

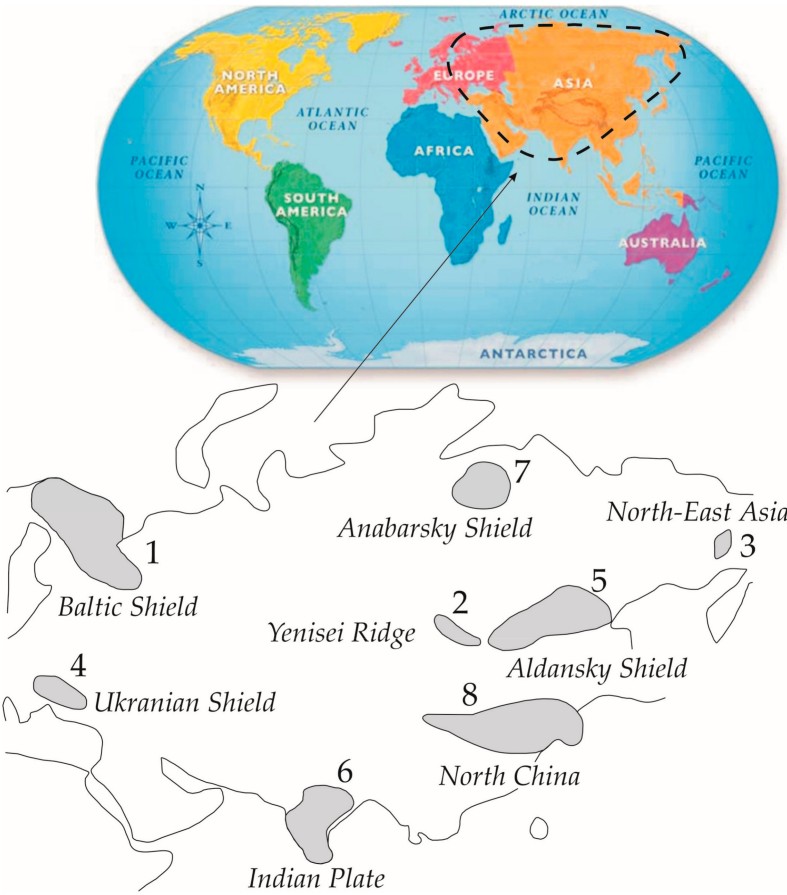

**Figure 2.** Scheme of the location of the Precambrian structures in Eurasia with the studied ancient collision belts.

Based on the above ideas, we suggest that the products of mantle magmatism within these structures should have similar petrogeochemical features, regardless of the specific nature of each region. If only the mantle composition in any of the listed regions is region-specific, this may be reflected in the composition of the mantle products within the magmatic complexes that formed in the region. Thus, data on the composition of the granulites in the Baikal region (Figures 2 and 3, point 2) have not been used for comparison because of the widespread Neoproterozoic–Paleozoic alkaline complexes that are exposed, in particular, among the rocks of the granulite complex (e.g., the Zhidoyskiy (Zadoyskiy) massif) [10].

Thus, the unique characteristics of the Kola region suggest that the early Precambrian proto-island arcs of the Lapland Granulite Belt could have specific features, with general patterns of chemical composition in all such structures in Eurasia. As we studied the formation of the Paleozoic Kola alkaline province, alkaline rocks typically contain highly alkaline indicator minerals, which are reflected in the increased contents of $Na_2O$ and $K_2O$ with commonly low contents of $SiO_2$ and/or $Al_2O_3$ in the rock compositions [5]. Most ultrabasic–basic alkaline rocks are known to be undersaturated in alumina and/or silica compared with alkalis. Therefore, we mainly focused on the distribution of these elements in early Precambrian complexes.

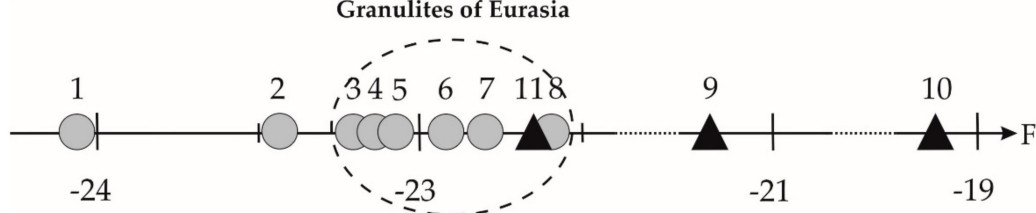

**Figure 3.** Location of points indicating compositions of rocks in the Kola region and granulite belts in Eurasia on the axis described by linear function $F = -0.39 \times SiO_2 - 0.37 \times TiO_2 - 0.38 \times Al_2O_3 + 0.12 \times \Sigma Fe + 0.05 \times MgO - 0.13 \times CaO + 0.44 \times Na_2O + 0.57 \times K_2O$. 1–7, ultrabasic-basic granulites of: (1) northeastern Baltic Shield (Lapland–Kolvitsa granulite belt); (2) Baikal region, Olkhon, and Sharyzhalgay series; (3) Northeast Asia; (4) Ukrainian Shield (Pobuzhye); (5) southern and southwestern part of the Aldan Shield; (6) South and Southeast India; (7) Anabar Shield; (8) North China (1–8 correspond to the same in Figure 2). (9)–(11) ultrabasic-basic magmatic rocks of the Paleoproterozoic (9) and Paleozoic (10) Kola region, and an average composition of ultrabasic-basic rocks that formed Neoarchean proto-island arc belts, and later structures of the Kola region up to the Paleozoic (**11**).

The difference in chemical rock composition in the transition from the Lapland Granulite Belt to other granulite belts of Eurasia was described by searching for a trend of differences with regard to the partial order introduced by a researcher into a preset aggregate of data sets. This method has been applied and detailed by authors in other studies [11]. Provided below are amendments to the method in the applied mathematical model for completing the new tasks in our research. This model was the basis used to search for a trend to be described by a linear function of chemical rock compositions.

Suppose Y is an n-dimensional random variable that characterizes the chemical compositions of the rocks composing the Lapland–Kolvitsa granulite belt. Suppose $Z = \{Z_i\}$ represents a set of n-dimensional random variables that characterizes the chemical compositions of the other studied granulite belts in Eurasia (n is the number of parameters that characterize chemical compositions of the rocks). Random variable Y is represented by a sample of n-dimensional vectors $W = \{w_j | w_j \in R^n\}$. Hence, each random variable, $Z_i$, is represented by a sample of n-dimensional vectors $V_i = \{v_{ij} | v_{ij} \in R^n\}$. The partial order is represented by $\{W < V_i$ for all i$\}$.

As we need to find a linear trend, constructing a linear function of $F: R^n \rightarrow R$ is sufficient, which will be the mathematical model of this trend. The required linear function is $F(x) = (c,x)$, where $(c,x)$ is a scalar product of a vector $c \in R^n$ of unit length $|c| = 1$ for certainty, and a vector of function arguments $x \in R^n$. It is represented by some straight lines in the space Rn (i.e., in the space of the chemical rock composition features).

To model function F, it is enough to find an n-dimensional vector c of unit length for which the mean value of a scalar product set $Yc = \{(c, w_j)\}_j$ is lower than the mean value of scalar products $Zc_i = \{(c, v_{ij})\}_j$ for each set $\{V_i\}$. The sets Yc and $Zc_i$ represent the projections of vectors $\{w_j\}$ and $\{v_{ij}\}$, respectively, on a straight line preset by the vector c. Thus, the following ratios must be fulfilled: $M(Yc) < M(Zc_i)$ for each i, where $M(.)$ is a mean value of the sample. We suggest using median $Me(.)$ to estimate the mean value.

The Puri–Sen–Tamura rank-order test for the equality of means can be applied to verify the $M(Yc) < M(Zc_i)$ hypothesis [12,13]. The choice of the specified statistical criterion is defined by both its stability against the violation of the normality (or even unimodality) condition for the random variable distribution and against the availability of anomalous observations in the samples. This ensures a high reliability of the resulting statistic conclusions. This criterion was used by others in numerous studies (e.g., Kozlov et al. [11]), which showed a high efficiency in the processing of a large volume of geochemical data. Hence, the ratios $Me(Yc) < M(Zc_i)$ and $\Lambda(Yc,Zc_i) > \chi^2(\delta)$ were executed for each sample, $V_i$, where $\Lambda(Yc,Zc_i)$ is the value of the Puri–Sen–Tamura statistics for sets Yc and $Zc_i$, and $\chi^2(\delta)$ is the value of the $\chi^2$-inverse distribution for the selected significance level $\delta$. Usually, one of 0.05 or 0.01 is selected for $\delta$.

Thus, the modeling of function F involves the search for such an n-dimensional vector c of unit length, for which, at a chosen significance level, δ, the following conditions are satisfied:

$$Me(Yc) < Me(Zc_i) \qquad (1)$$

$$\Lambda(Yc,Zc_i) > \chi^2(\delta) \qquad (2)$$

where $\chi^2(\delta)$ denotes a value of the $\chi^2$-inverse distribution for the significance level, δ, of all pairs $<Yc,Zc_i>$.

Notably, the search for an n-dimensional vector c that fulfills the above criteria provides one of the following results. First, no vector c of the kind may exist. Second, the vectors that fulfill these criteria may form a set with one or more vectors. In the first case, the forced conclusion is that there is no required linear trend (linear function F(x) = (c,x)) in the transition from the Lapland–Kolvitsa granulite belt to the other granulite belts in Eurasia. Otherwise, the optimal solution must be selected from a set of probable solutions, which can be found by solving the following classical functional optimization task:

$$J(c) = \max_c \min_i \Lambda(Yc,Zc_i) \qquad (3)$$

with the following constraints:

$$Me(Yc) < Me(Zc_i) \qquad (4)$$

$$\Lambda(Yc,Zc_i) > \chi^2(\delta) \qquad (5)$$

$$|c| = 1 \ (6) \qquad (6)$$

For all pairs $<Yc,Zc_i>$, $c \in R^n$, $\chi^2(\delta)$ is a value of the $\chi^2$-inverse distribution for the significance level δ. Thus, the functional J(c) optimization means selecting such a vector (c) that, with the above-mentioned constraints, the minimal values of the Puri–Sen–Tamura statistics are obtained for all pairs of sets $<Yc,Zc_i>$.

The optimal solution (i.e., vector c) was searched using the Nelder–Mead algorithm [14].

## 4. Results

The task was formulated to search for a linear trend in the chemical rock compositions that satisfies the following condition. The primarily magmatic ultrabasic-basic rocks of the Lapland Granulite Belt should, against this trend, have a statistically significant difference from these rock formations (in terms of geochemical classification) of the other belts in Eurasia. This linear trend was found; it is represented by the linear function F(x) for the parameters of the chemical rock composition (Figure 3, and Tables 1 and 2). Here, the difference in the projections of the chemical composition of this trend of the Lapland Granulite Belt rocks in the transition to the rock assemblages of the other studied complexes is the reduced content of alkalis at the simultaneous general increase in the content of $SiO_2$ and $Al_2O_3$. Importantly, the increase or decrease in an element with regard to this trend means the trends describe the projections of the chemical rock composition on the obtained trend. However, the absolute contents of various elements may insignificantly vary. In this case, only the behaviors of the $SiO_2$, $Al_2O_3$, and total alkalis are particularly important in this research, not that of all petrogenic elements. Notably, the trends revealed for these features persist in their comparison. Thus, a tendency toward an increase in the contents of $SiO_2$ and $Al_2O_3$ (49.74% and 15.78% vs. 48.91% and 14.32%, respectively) is suggested in the rocks of the Lapland Granulite Belt in comparison with the metamorphic rocks of the other granulite belts in Eurasia (except for the Baikal region). We found a slight trend toward a reduction in the total alkalis (3.04% vs. 3.10%), at a noticeable reduction in $K_2O$ content (0.54% to 0.68%) in the rocks of the Kola region (Table 2).

**Table 1.** Medians (highlighted in bold) and values of the Puri–Sen–Tamura statistics for function F for the Lapland Granulite Belt and granulites of Eurasia.

| | 1 * | 2 | 3 | 4 | 5 | 6 | 7 | 8 |
|---|---|---|---|---|---|---|---|---|
| 1 | **−24.0538** | | | | | | | |
| 2 | 14.2868 ** | **−23.4629** | | | | | | |
| 3 | 29.8476 | 4.4156 | **−23.2626** | | | | | |
| 4 | 15.4656 | 3.2608 | 0.0421 | **−23.1697** | | | | |
| 5 | 76.8176 | 6.5185 | 0.2001 | 0.3827 | **−23.1135** | | | |
| 6 | 32.1595 | 5.1038 | 0.0020 | 0.0027 | 0.3361 | **−22.9846** | | |
| 7 | 20.8434 | 4.9224 | 0.2633 | 0.0120 | 1.0685 | 0.3143 | **−22.8459** | |
| 8 | 15.4354 | 4.4184 | 0.4969 | 0.3188 | 1.6356 | 0.5567 | 0.3773 | **−22.6565** |

Note: * figures indicate rock complexes (Figure 3); ** values of the Puri–Sen–Tamura statistics more than 3.8 for 95% probability of differences, and more than 6.6 for 98%.

**Table 2.** The average composition (%) of the studied basic-ultrabasic rock groups in the Kola region and the granulite belts of Eurasia.

| | $SiO_2$ | $TiO_2$ | $Al_2O_3$ | $Fe_2O_3$ | FeO | MnO | MgO | CaO | $Na_2O$ | $K_2O$ |
|---|---|---|---|---|---|---|---|---|---|---|
| 1 * | 49.74 | 1.05 | 15.78 | 2.41 | 8.98 | 1.83 | 7.18 | 10.16 | 2.50 | 0.54 |
| 2 | 48.49 | 1.37 | 14.56 | 3.66 | 8.70 | 0.20 | 7.17 | 11.83 | 2.42 | 0.50 |
| 3 | 48.95 | 1.13 | 14.30 | 4.02 | 8.24 | 0.21 | 8.38 | 9.25 | 2.35 | 1.00 |
| 4 | 48.02 | 1.41 | 14.52 | 3.65 | 10.82 | 0.20 | 7.31 | 10.38 | 2.01 | 0.52 |
| 5 | 48.89 | 1.16 | 14.40 | 4.07 | 8.83 | 0.22 | 7.32 | 10.64 | 2.43 | 0.62 |
| 6 | 49.39 | 1.27 | 13.89 | 2.13 | 11.36 | 0.30 | 7.56 | 10.10 | 2.44 | 0.69 |
| 7 | 48.50 | 1.32 | 14.73 | 6.16 | 6.88 | 0.20 | 6.96 | 10.22 | 2.71 | 0.80 |
| 8 | 49.10 | 0.65 | 13.11 | 4.00 | 8.23 | 0.18 | 9.82 | 9.56 | 2.50 | 0.95 |
| 9 | 49.04 | 1.51 | 15.73 | 3.87 | 6.11 | 0.21 | 4.97 | 6.77 | 5.57 | 3.05 |
| 10 | 49.51 | 1.51 | 19.50 | 3.72 | 2.86 | 0.24 | 1.40 | 3.12 | 9.23 | 6.64 |
| 11 | 49.56 | 1.21 | 16.29 | 2.90 | 7.52 | 1.27 | 5.91 | 8.46 | 4.09 | 1.92 |
| 12 | 48.91 | 1.16 | 14.32 | 3.96 | 8.94 | 0.22 | 7.52 | 10.39 | 2.42 | 0.68 |

Note: * Numbers of groups correspond to the numbers in Figure 3. Group 12 denotes the average rock composition of the Eurasian granulite belts (except for the Lapland granulite belt and Baikal region).

As stated above, a peculiar feature of the Paleozoic alkaline rocks of the region is an increased content of alkaline elements with a specific behavior of $SiO_2$ and/or $Al_2O_3$, which is opposed to the behavior of these elements described by a linear trend F.

## 5. Discussion

We suggest different reasons for the specific composition of the Archean mantle products in the Kola region compared with similar rocks in the other regions where alkaline complexes are less widespread. First, a mantle with a specific composition generated ultrabasic-basic rocks of the Lapland–Kolvitsa granulite belt. The mantle was initially poor in alkaline elements and rich in $SiO_2$ and $Al_2O_3$. This suggestion is rather doubtful, as such a mantle could not be a source for the later products of magmatic activity that are characterized by the diametrically opposite behavior of every listed element.

Second, the composition can be explained by the melting of products out of the mantle, which is considered normal for such areas in the Earth. These products are alkali-poor and $SiO_2$- and $Al_2O_3$-rich

during the first stages during differentiation. In this case, residual mantle products can be expected to be rich in alkalis and poor in $SiO_2$ and $Al_2O_3$. This does not contradict the petrogeochemical features of Paleozoic rock complexes in the region, and therefore seems logical, for example, the extraction of melts is poor in alkaline elements and rich in $SiO_2$ and $Al_2O_3$ [15].

The second explanation seems more logical. If this is correct, it is possible, considering the above ideas and with no mathematic differentiation, that the petrogeochemical features of the hypothetic rocks of the mantle genesis in the Kola region should have been similar to the same formations of the other studied objects in Eurasia. The composition of these hypothetic rocks, which could have formed in the Kola region as a result of a single magmatic event with no differentiation processes in the mantle, can be derived from the compositions of all of the ultrabasic-basic rocks in the abovementioned Neoarchean, Paleoproterozoic, and Paleozoic complexes. For this purpose, they were considered in proportions of 65%, 21%, and 14%, respectively. This reflects the approximate proportion of their distribution in the region (present estimations [2] for Paleozoic rocks of the Kola Alkaline Province). This task was addressed using the data on the compositions of ultrabasic-basic magmatism products of all of the structures mentioned in Section 2 (about 1400 bulk rock chemical analyses). The details on the average compositions of the studied rock assemblages are listed in Table 2. The samples of the Paleoproterozoic metamorphosed igneous rocks were expanded by metabasalts and metapicrobasalts of the Pechenga, Imandra–Varzuga, and Pana–Kuolayarvi structures, as well as by the basic-ultrabasic igneous rocks of the Monche and Pana Tundras. The Paleozoic rock sample includes mafic–ultramafic igneous rocks of the alkaline ultrabasic series of the Lovozero, Khibiny, and Kovdor massifs.

The point of the average composition of these hypothetic rocks with the mantle origin on the axis represented by linear function F(x) is visibly displaced toward the compositions of the basic magmatism products in all of the studied granulite belts in Eurasia. This finding can be easily explained if the position of the compositions of the Paleoproterozoic and Paleozoic formations is considered to lie on this axis (Figure 3). Thus, this finding corroborates the previous suggestion that mantle products and, indirectly, the mantle, have similar compositions in areas with close geodynamic settings.

## 6. Conclusions

The petrogeochemical modeling suggests that the earlier differentiation of the subcrustal substrate participated in the formation of the Kola alkaline province as well. Thus, these data show that the unique early Precambrian volcanism in this region may be one of the reasons for the unique composition of products of Paleozoic magmatism in the northeastern Baltic Shield. The conclusion about the early Precambrian history of the Kola alkaline province formation may be more universal, indicating a tight link between the late and younger geological events. From the granulites of the Baikal region also being characterized by widespread Neoproterozoic and Paleozoic alkaline complexes, it can be inferred that factor F (Figure 3, Table 1) is closely related to the granulites of the Kola region, and differs from similar structures in Eurasia. In this case, the data provided here can be used by experts in the metallogenic analysis of ancient platforms.

These assumptions require further study of the reasons for the specific evolution of the Early Precambrian proto-island arc in the Kola collision area. We do not speculate about the mechanism for the accumulation of alkalis and the primarily incoherent potassium in the mantle. As an option, the concentration of K and Na by double carbonates at depths of 180 to 210 km and temperatures of below 1200 and 1300 °C could be considered [16]. This issue, however, was beyond the scope of the current study and remains disputable.

**Author Contributions:** N.E.K. conceived the research, designed the experiments, and prepared the main part of the manuscript. N.O.S. provided petrographic research, and jointly wrote the Discussion section with N.E.K. E.V.M. elaborated methods of data processing, applied them to the current research, and wrote the part dedicated to their description. All of the authors discussed and approved the manuscript. All authors have read and agreed to the published version of the manuscript.

**Funding:** The research leading to this paper was funded by the Geological Institute of the Kola Science Centre RAS, and the P.P. Shirshov Institute of Oceanology RAS, public contracts no. 0149-2019-0005 and no. 0226-2019-0052.

**Acknowledgments:** The authors thank T.S. Marchuk for her professional and high-quality processing of the geochemical data and the design of this paper. We thank the anonymous reviewers for constructive comments and valuable suggestions for improvement.

**Conflicts of Interest:** The authors declare no conflict of interest. The founding sponsors had no role in the design of the study; in the collection, analyses, or interpretation of data; in the writing of the manuscript; and in the decision to publish the results.

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
