# Peer review of "Using Petrogeochemical Modeling to Understand the Relationship between Paleozoic Magmatism in the Kola Region and Its Precambrian History"

_geosciences, doi:10.3390/geosciences10010011_

Round 1

Reviewer 1 Report

In the Abstract it is necessary to mention the ages of individual magmatic complexes, that is, what is Palaeozoic and what Precambrian is. – lines 11-14. Please, do not use abbreviations without prior explanation the full name, see PGE, probably "platinum group minerals" (line 36). In Figure 1 you have the discorded legend. On the one hand you have the "Meso - and Neoproterozoic sediments" and then "Meso - and Neoarchean rocks" (which type) and "the Paleoproterozoic rift basins" (basin filling?). It should be also in the legend in more detail to specify. Please add in the Introduction of citations of the articles from which were drawn the original chemical analysis of the granulite areas of the Euroasia. Also need to fix the number to the quotation in line 39. Citation number 13 corresponds to article Tamura (1966). Line 73-74 – Please, be sure to include method of dating. Line 117, Fig. 2: Scheme of the location of Precambrian structures in Eurasia with the studied ancient collision belts. I think this map is too schematic. It would be good to add at least coordinate system. Note: The use of the initials first name in the quotation of authors in the text, it is not necessary, e.g. A. A. Arzamastsev, L. N. Kogarko. Please fix it. Conclusion it is clearly needs to be reviewed. As is done means continued chapter Discussion. It must be done briefly and expressively, directed on the main results of the research referred to in the article. The list of references is necessary to edit within the meaning of the regulations of the journal Geosciences.

Author Response

Dear Editor(s),

please see the attachment!

Reviewer 2 Report

The study presents some interesting results, and appears to fill an important hole in the literature. My primary concern is with the writing style. The manuscript could use some considerable re-structuring and editing of the english. The phrasing is often awkward, and commonly hides the meaning, and significance, of the work. This can often make it difficult to understand what the author's are trying to communicate, and how their work relates with the previous work from the region.

The writing is commonly verbose, and the fragmented sentence structure can be very difficult to follow. There are some entire paragraphs that are unnecessary (for example: the first paragraph in the discussion, which doesn't seem to add any information, and seems to be 'drawing a conclusion' before the results have been discussed).

Some words have been incorrectly used (for instance, the pluaralisation of research as 'researches', same with 'evidences'). "Ga" is also incorrectly used in Line 84 - Ga should denote 'billions of years ago', whereas Gy is prefered for 'billions of years' (i.e. the time between two events).

There's quite a few places where there's unnecessary repetition (for instance, Line 59-60 "As it was mentioned above, the geology of the Kola alkaline province is described in detail by A.A. Arzamastsev and coauthors" - if it's mentioned above (and it's not clear that it is), then it doesn't need to be said again.

There are also several places where the author's imply that the study is unnecessary or unimportant (for instance: Line 39: " has been examined in sufficient detail" -  to me this sentence implies that no further work is needed (so why this study?). It may have been extensively studied, but I'm sure there's more work to be done). The author's actually do present some novel results, so I think they need to be careful with their phrasing.

There are some vague references, for instance Line 73 "magmatism manifested within the province is dated accurately enough" - I think it'd be helpful to say how it was dated here. 380-360 million years is a considerable time span. Another good example is Lines 252-254 - how can 'SiO2, CaO and Al2O3' increase, without addition to the system? Do you mean they are increasing in the melt (while decreasing in the residual mantle). Sentences like this make the argument very hard to follow.

This paper should be re-structured and re-written to present the results more clearly. I would suggest getting a native english speaker to help prepare a revised draft.

Author Response

(The authors gave the same response as above.)
